# Maternal Sepsis in Italy: A Prospective, Population-Based Cohort and Nested Case-Control Study

**DOI:** 10.3390/microorganisms11010105

**Published:** 2022-12-31

**Authors:** Sara Ornaghi, Alice Maraschini, Marta Buoncristiano, Edoardo Corsi Decenti, Elisabetta Colciago, Irene Cetin, Serena Donati

**Affiliations:** 1Department of Obstetrics, MBBM Foundation Onlus at San Gerardo Hospital, 20900 Monza, Italy; 2School of Medicine and Surgery, University of Milan-Bicocca, 20900 Monza, Italy; 3Technical-Scientific Statistical Service, Istituto Superiore di Sanità—Italian National Institute of Health, 00161 Rome, Italy; 4National Centre for Disease Prevention and Health Promotion, Istituto Superiore di Sanità—Italian National Institute of Health, 00161 Rome, Italy; 5Department of Biomedicine and Prevention, University of Rome Tor Vergata, 00133 Rome, Italy; 6Department of Biomedical and Clinical Sciences, University of Milan, 20122 Milan, Italy; 7Department of Obstetrics and Gynecology, V. Buzzi Hospital, ASST Fatebenefratelli Sacco, 20154 Milan, Italy

**Keywords:** sepsis, infection, organ failure, peripartum, near miss, pregnancy, delivery

## Abstract

Maternal sepsis represents a leading cause of mortality and severe morbidity worldwide. In Italy, it is the second cause of direct maternal mortality. Delay in recognition and treatment initiation are the drivers of sepsis-associated adverse outcomes. Between November 2017 and October 2019, the Italian Obstetric Surveillance System coordinated a prospective population-based study on maternal sepsis occurring before or after childbirth from 22 weeks’ gestation onward and up to 42 days following the end of pregnancy. A nested 1:2 matched case-control study on postpartum sepsis was also performed. Maternal sepsis was diagnosed for the presence of suspected or confirmed infection alongside signs or symptoms of organ failure. The aim of this study was to assess maternal sepsis incidence and its associated risk factors, management, and perinatal outcomes. Six Italian regions, covering 48.2% of the national births, participated in the project. We identified an incidence rate of 5.5 per 10,000 maternities (95% CI 4.80–6.28). Seventy percent of patients had a low education level and one third were foreigners with a language barrier. Genital, respiratory, and urinary tract infections were the predominant sources of infection; the majority of cases was caused by *E. coli* and polymicrobial infections. The presence of vascular and indwelling bladder catheters was associated with a nine-fold increased risk of postpartum sepsis. There were no maternal deaths, but one fourth of women experienced a serious adverse event and 28.3% required intensive care; 1.8% of newborns died. Targeted interventions to increase awareness of maternal sepsis and its risk factors and management should be promoted.

## 1. Introduction

Strategic approaches to reduce maternal mortality in the last two decades have mainly focused on clinical interventions and health system strengthening [1]. The greatest attention has initially been on postpartum hemorrhage and hypertensive disorders, the two leading causes of direct maternal mortality worldwide [2]. Further initiatives have then been developed to address maternal sepsis [3], which represents the third most common direct cause of maternal mortality, by being responsible for almost 11% of all maternal deaths [2,4,5,6].

Adequate assessment of the burden associated with maternal sepsis has been challenging, due to the lack of a clear, evidence-based, and actionable definition of the condition [7]. This has led the World Health Organization (WHO) to launch the Global Maternal Sepsis and Neonatal Initiative in 2016, alongside convening an expert consultation for developing a new, shared definition of maternal sepsis [8], which reflected the thinking embedded in the 2016 *Third International Consensus Definitions for Sepsis and Septic Shock* (SEPSIS-3) [9].

Undetected or poorly managed maternal infections can lead to sepsis, and, in turn, disability and death for the mother and the increased likelihood of early neonatal infection and additional adverse outcomes [3]. Thus, maternal sepsis can be considered a “near-miss” event [10]. “Near-miss” events are proxies of maternal health care quality, and their monitoring and in-depth investigation provide an essential feedback to improve obstetric care [11].

The Italian Obstetric Surveillance System (ItOSS) coordinates an enhanced maternal mortality surveillance system [12,13] and participates in multi-country population-based prospective studies with the aim to investigate uncommon severe maternal morbidity, in collaboration with the International Network of Obstetric Survey Systems (INOSS) [14]. According to the ItOSS retrospective record-linkage procedures of vital statistics, maternal sepsis was the fourth cause of direct maternal mortality in 10 Italian regions covering 77% of national births between 2006 and 2012, with a specific maternal mortality ratio of 0.2 per 100,000 live births [13]. In 2013–2017, the incident reporting and confidential enquiries into maternal deaths coordinated by the ItOSS surveillance identified maternal sepsis as the second cause of direct maternal deaths, being responsible of 23 cases (20.8%), and only following obstetric hemorrhage (22 cases, 21.7%) [12].

Given the substantial relevance of maternal sepsis in maternal morbidity and mortality in Italy [12], where there is a paucity of high-quality studies on this condition, ItOSS implemented a prospective population-based study on maternal “near-miss” events, including maternal sepsis [15,16].

The aim of this study was to estimate the incidence of maternal sepsis, to investigate risk factors and management of this condition, and to describe the associated maternal and perinatal outcomes.

## 2. Materials and Methods

### 2.1. Study Design

The present study is part of a larger prospective population-based cohort study of obstetric “near-misses” conducted in Italy by ItOSS [17]. A maternal “near-miss” case is defined as “a woman who nearly died but survived a complication that occurred during pregnancy, childbirth or within the 42 days of termination of pregnancy” [11]. ItOSS collected data on women with any of the following events: sepsis, eclampsia [16], amniotic fluid embolism, and spontaneous hemoperitoneum in pregnancy [15]. This manuscript focuses on sepsis.

### 2.2. Setting

The ItOSS study originally involved nine regions (Piedmont, Lombardy, Friuli-Venezia Giulia, Emilia-Romagna, Tuscany, Latium, Campania, Apulia, and Sicily) of Italy. A few months after the start of the study, monthly checking of data reporting by ItOSS allowed for the identification of substantial underreporting and missing data for Latium, Campania, and Apulia; this led the ItOSS investigators to decide to exclude such regions from the study to avoid biased reporting. Thus, data on maternal sepsis has been collected in six Italian regions, located in the north (Piedmont, Lombardy, Friuli-Venezia Giulia, and Emilia-Romagna), center (Tuscany) and south (Sicily) of Italy, covering 48.2% of total births in the country. A total of 200 public and private maternity units were included; each unit had a nominated clinician responsible for case reporting. Data collection lasted 22 months, from 1 November 2017 until 31 October 2019.

### 2.3. Participants

Maternal sepsis was defined as a life-threatening condition determined by organ failure resulting from infection during pregnancy, childbirth, or up to 42 days post-abortion or postpartum, in line with the definition adopted into the Surviving Sepsis Campaign guidelines and with the thinking of the SEPSIS-3 Consensus [3,7,9]. Of note, parameters for sepsis diagnosis developed during this consensus are not only limited by a low positive predictive value but also by the lack of validation in the obstetric population. For this reason, the WHO promoted the multicenter Global Maternal Sepsis Study (GLOSS) in 2017, aiming to define and validate a set of criteria for identifying cases of suspected or certain maternal sepsis [5].

While awaiting the GLOSS’ results, ItOSS has convened a multidisciplinary panel of experts and defined a set of criteria for the diagnosis of organ failure and infection to be employed in the ItOSS project [18], specifically considering the physiological changes of vital parameters occurring during pregnancy and puerperium [19].

The diagnosis of organ failure was based on one or more of the following vital signs or laboratory exams: (1) systolic blood pressure < 90 mmHg or mean arterial pressure < 65 mmHg (cardiovascular criteria), (2) need of oxygen to maintain SpO2 > 92–93% (respiratory criteria), (3) creatinine > 1.2 mg/dL or an increase > 0.5 mg/dL and/or diuresis < 0.5 mL/Kg for >2 h (renal criteria), (4) bilirubin ≥ 1.2 mg/dL (liver criteria), (5) altered state of consciousness by means of Alert, Vigilant, Pain, Unconscious assessment scale (central nervous system criteria) [20], and (6) platelets < 100.000/mm^3^ or a decrease of ≥50% compared to usual levels in pregnancy (hematology criteria). Diagnosis of infection was based on at least one of the following signs or symptoms: body temperature > 38 °C or < 35 °C, headache or nuchal rigidity, respiratory symptoms (productive cough, pharyngodynia, etc.), breathing difficulty (frequency ≥ 22 respiratory acts/min; SpO2 < 95%), urinary symptoms (dysuria, cloudy urine, etc.), abdominal or pelvic pain, diarrhea or vomiting, skin rash, foul-smelling vaginal discharge, threatening preterm labor or preterm premature rupture of membranes (pPROM), foul-smelling or purulent amniotic fluid in case of pPROM, and fetal or neonatal signs of infection.

ItOSS differentiated cases of sepsis occurred in case of miscarriage, ectopic pregnancy, termination of pregnancy, or molar pregnancy up to 21^6/7^ weeks of gestation (i.e., antepartum sepsis) from those diagnosed either before or after childbirth from 22 weeks’ gestation onward and up to 42 days following the end of pregnancy (i.e., peripartum cases).

This manuscript presents the analyses regarding women with peripartum sepsis who gave birth vaginally or by cesarean section (CS) at or after 22 weeks’ gestation in the participating maternity units. In addition, findings of a 1:2 matched case-control study including only cases with a diagnosis of sepsis after childbirth are presented herein. Clinicians who reported such cases of sepsis were asked to identify two non-septic women who delivered in the same hospital immediately before the index case and by the same mode, and also to complete an anonymized data collection form.

### 2.4. Data Sources

As previously described [15,16], a reference clinician from each maternity unit was identified to attend a single-day training course at regional level to share the study’s objectives and methodology and learn how to use the web platform for data entry.

Incident cases were reported using electronic and anonymous data collection forms including information regarding the women’s sociodemographic characteristics, medical and obstetric history, timing and mode of delivery, detailed description of the event and the employed treatment, clinical and instrumental diagnosis, source of infection, causative pathogens, and maternal and perinatal outcomes. Maternal mortality, admission to intensive care unit (ICU), and severe maternal morbidities were investigated as maternal outcomes. Perinatal mortality, admission to neonatal intensive care unit (NICU), and major neonatal complications were analyzed as perinatal outcomes.

Each reference clinician received a monthly reminder to promote complete reporting or to confirm the null report.

### 2.5. Statistical Analyses

The incidence rate was calculated as the number of sepsis cases per 10,000 maternities with 95% confidence interval (CI), assuming the Poisson approximation to the binomial distribution. Descriptive analyses of cases’ characteristics, childbirth features, causative organisms, sources of infection, and perinatal outcomes were performed. Also, to assess risk factors for developing postpartum sepsis, all cases diagnosed with sepsis after childbirth were compared with non-septic controls in a 1:2 matched case-control fashion. A conditional logistic regression (CLR) model, a specialized type of logistic regression usually employed when case subjects with a particular condition are each matched with *n* control subjects without the condition, was applied, accounting for matching factors. The adjusted odd ratios (aOR) with 95% Confidence Intervals (CI) for maternal postpartum sepsis were calculated for each variable of interest.

### 2.6. Missing Data Were Reported in the Tables

Data analyses were performed at the Italian National Institute of Health using Stata/MP 14.2 (Stata Corp., College Station, TX, USA).

### 2.7. Ethical Approval

The project was approved by the Ethics Committee of Italian National Institute of Health (Prot. PRE-544/17, Rome) on 18 July 2017. Data were fully anonymized before being accessed and analyzed. Thus, the need for informed consent was waived by the local Ethics Committee.

## 3. Results

Ten of the 200 maternity units in the six selected regions did not provide the requested data, thus leading to an overall participation rate of 95%.

From 1 November 2017 until 31 October 2019, 355 cases of maternal sepsis were notified. Twenty-three cases were excluded: nine had missing information regarding the gestational age at diagnosis of sepsis, five were reported from Latium, Campania, and Apulia after their formal exclusion, three were duplicates, and six occurred outside the study time frame, thus leading to 332 cases. Antepartum sepsis occurring before 22 weeks’ gestation was diagnosed in 113 (34.0%) women, whereas the remaining 219 (66.0%) women were identified as cases of peripartum sepsis and, thus, were included in the present analyses.

During the study period, 398,202 women gave birth in the six participating regions, for an overall estimated incidence rate of peripartum sepsis of 5.5 per 10,000 maternities (95% CI 4.80–6.28).

Among the 219 women with peripartum sepsis, 212 (96.8%) had a livebirth, whereas seven had a stillbirth. Maternal sepsis was identified after childbirth in 166 (75.8%) cases (three missing): 165 livebirths and one stillbirth.

The cardiovascular as well as the respiratory criteria for identifying organ dysfunction were the most common, with 49.8% and 36.1% of women presenting them, respectively (Figure 1). In turn, the additional criteria, such as renal or liver abnormalities, were recognized in less than 15% of women each, with the altered state of consciousness being the less common identified criterion (n = 16 women, 7.3%).

Table 1 displays the demographic characteristics and the medical and obstetric history of peripartum sepsis cases.

Approximately one third of women had an advanced age ≥ 35 years, was a foreigner, and showed difficulty in communication due to a language barrier. A low education level was identified in almost 70% of patients. The pregestational body mass index was underweight in 5.5% of cases and obese in 13.2% of them. A history of recurrent infections was reported in 13.2% of patients; furthermore, three patients were diagnosed as immunocompromised (high-dose corticosteroid therapy). Almost 60% of women were nulliparas; 8.2% of pregnancies were conceived artificially, and in 33.3% of these gestations heterologous techniques were employed. Approximately one fourth of patients were undergoing antibiotic therapy during the two weeks preceding the diagnosis of sepsis.

Details regarding the childbirth event of enrolled women are shown in Table 2.

Half of women who experienced peripartum sepsis labored: 28.3% of them had an induction of labor and 49.5% required epidural analgesia. Vaginal prostaglandins were the most common method of labor induction, being employed in 67.7% of cases, followed by oxytocin in 33.9% of cases, artificial rupture of membranes in 17.7% of cases, and mechanical methods (e.g., cervical ripening balloon or Foley catheter) in 12.9% of cases. Grade 2- or 3-stained amniotic fluid was identified in 9.6% of cases; almost one fourth of women received more than five vaginal examinations during labor.

Sixty-three (28.8%) births occurred prematurely, seven of which <28^0/7^ weeks’ gestation, with higher rates among cases of sepsis diagnosed antepartum compared to those postpartum (56% versus 22%, *p* < 0.001).

CS was the mode of birth in 156 (71.2%) women, and in 83.3% of them it was classified as urgent or emergent. There were no cases of either incomplete manual placental removal or dilation and curettage. Forty-three (19.6%) patients experienced postpartum hemorrhage >500 mL, and in two cases a peripartum hysterectomy was required.

Abdominal or pelvic pain was the most common presenting symptom, being identified in 33.3% of women (Figure 2A). One third of cases complained of flu-like symptoms and, in line with this observation, respiratory signs, such as productive cough or pharyngodynia, were present in 23.3% of septic patients. Signs of urinary discomfort, including dysuria and cloudy urine, were present in 17.4% of women. Fever with body temperature > 38 °C was diagnosed in 73.5% of patients, followed by tachycardia > 100 bpm in 45.7% (Figure 2B); in turn, less than one third of cases displayed leukocytosis >17 × 10^9^/L.

Seventeen (7.8%) women did not receive any diagnostic imaging evaluation to identify the source of infection; in addition, serum lactate levels were not evaluated in 30.6% of cases. Among patients who delivered by CS (n = 156), all but two (1.3%) received antibiotic prophylaxis, whereas this was administered in 27 out of 63 cases of vaginal delivery (n = 7 for vagino-rectal swab positive for group B Streptococcus, GBS, n = 2 for unknown result of vagino-rectal swab for GBS, n = 4 for PROM > 18 h, and n = 14 for attending clinician’s decision).

The sources of infection in the overall study population are shown in Figure 3, whereas Table 3 displays the sources of infection according to the timing of sepsis diagnosis, i.e., before or after childbirth, and mode of birth, i.e., vaginal delivery or CS, respectively.

The source of infection was identified in 81.7% of women. The genital tract, the respiratory tract, and the urinary tract were the most common sources, being involved in 26%, 17.8%, and 16% of cases, respectively (Figure 3).

Information regarding the timing of sepsis diagnosis was missing in three (1.4%) women. Respiratory and urinary tract infections were the most common sources of infection in women with an antepartum sepsis diagnosis, whereas genital tract infection was the most frequent source among women with a postpartum diagnosis (Table 3). Patients with postpartum sepsis also displayed higher rates of skin infections compared to antepartum sepsis patients (7.8% versus 0%, *p* = 0.042). When analysis was performed according to the mode of birth, we identified genital and urinary tract infections as the most frequent sources in women who delivered vaginally; in turn, women with CS more commonly showed infectious involvement of the genital and the respiratory tracts (Table 3).

Infection was confirmed by the laboratory (i.e., positive culture) in 67. 6% of women. *Escherichia coli* (*E. coli*) was the most frequently isolated germ among women with positive culture (n = 46/148, 31.1%); polymicrobial infections were the second most common type of infection (n = 33, 22.3%), followed by *Staphylococcus* (n = 21, 14.2%) (Figure 4A). Figure 4B–F displays the distribution of identified pathogens among the different sources of infection in laboratory-confirmed sepsis cases (n = 25 missing). Genital and urinary tract infections had *E. coli* as the most frequent isolated germ, skin infections had *Staphylococcus* as the most frequent isolated germ, and respiratory tract and other organ infections had polymicrobial infections as the most frequent isolated germ.

Among women with negative cultures (n = 71, 32.4%), 22 (31%) had an unknown source of infection.

The types of Staphylococcus isolated included the following: Staphylococcus aureus, methicillin resistant Staphylococcus aureus (MRSA), Staphylococcus coagulase negative, Staphylococcus capitis, Staphylococcus epidermidis, Staphylococcus hominis, Staphylococcus lugdunensis, and Staphylococcus warneri.

Antibiotic treatment was administered in 99.1% of Septic Women. Information regarding the timing of antibiotic therapy initiation was available in 175 (79.9%) cases: sixty-two (35.4%) received antibiotics before the diagnosis of sepsis, 44 (25.1%) received antibiotics within one hour from diagnosis, and 69 (39.4%) received antibiotics at or after one hour from diagnosis, with 16 (9.1%) being treated within 1 to 3 h and the remaining 53 (30.3%) being treated at more than 3 h.

One-hundred and fifty (69.2%) women received two or more antibiotics, primarily penicillin (146/490 overall antibiotic administrations, 29.8%) (Figure 5A,B).

Hypotension was identified in 148 (67.6%) patients. Volume resuscitation was performed in 52% of women with evidence of hypotension, and crystalloids at 30 mL/Kg were employed in 90.9% of these cases. Furthermore, vasopressor agents were administered in fourteen out of 148 hypotensive women.

A serious adverse maternal complication occurred in 51 (23.3%) women, with need of ventilation (n = 12, 23.5%), pulmonary edema (n = 11, 21.6%), and acute respiratory distress syndrome (n = 11, 21.6%) being the most frequent conditions. Renal insufficiency as well as HELLP syndrome were diagnosed in six cases each (11.8%); five (9.8%) women experienced a thrombotic complication, whereas only one had disseminated intravascular coagulopathy. Sixty-two (28.3%) women were admitted to the ICU: they were more frequently ≥35 years of age (45.2% versus 25.8%, *p* = 0.009) and multiparas (58.1% versus 34.2%, *p* = 0.002), and had more commonly received an intrauterine balloon tamponade before the diagnosis of sepsis (9.7% versus 0.6%, *p* = 0.002). There were no cases of cerebrovascular events or cardiac arrest among the study cohort; in addition, no maternal deaths were identified.

Information regarding neonatal outcomes was missing for five out of the 223 livebirth neonates. The median birthweight was 3065 g (min-max, 544–4360 g) among livebirth newborns (n = 218) and 650 g (min-max, 455–2050 g) among stillbirth infants (n = 7). Among the 218 livebirth neonates with available data, there were 11 (5.0%) and 5 (2.3%) cases with an Apgar score at 5 min < 7 and umbilical cord pH < 7.0, respectively. Sixty-eight (31.2%) newborns were admitted to the NICU, whereas death occurred in four cases (1.8%).

The nested case-control study in women with postpartum sepsis included 150 cases (n = 16 excluded for lack of data in the variables of interest) and 297 controls (n = 3 women with only one control reported). The results of the CLR model with adjusted odds-ratios are displayed in Table 4. The presence of vascular and indwelling bladder catheters were the only factors identified as associated with increased odds of postpartum sepsis (aOR 9.02, 95% CI, 1.53–53.09 and aOR 9.86, 95% CI 2.00–48.62, respectively); in turn, nulliparity, prior recurrent infections, pregestational medical conditions, and PROM only approached statistical significance (aOR 1.76, 95% CI 0.94–3.26; aOR 2.68, 95% CI 0.99–7.27; aOR 1.93, 95% CI 1.00–3.69; aOR 3.08, 95% CI 0.99–9.53, respectively).

## 4. Discussion

This study showed that the incidence of maternal sepsis in the participating Italian regions during the investigated time period was 5.5 per 10,000 maternities.

Genital, respiratory, and urinary tract infections were the predominant sources of infection, with substantial differences according to the timing of sepsis diagnosis and the mode of birth. The largest proportion of cases of sepsis was caused by *E. coli* and polymicrobial infections, and the presence of vascular and indwelling bladder catheters were significantly associated with the increased odds of postpartum sepsis.

Prior population-based studies of maternal sepsis have reported an incidence rate ranging from 2.1 per 10,000 maternities (The Netherlands, 2004–2006, and Scotland, 1986–2009) [21,22] to 4.7 (United Kingdom, 2011–2012) [23]—4.9 per 10,000 maternities (United States, 2005–2007) [24]. A more recent Canadian population-based study investigating cases of maternal sepsis was undertaken in the country between 2004 and 2017, and identified a higher incidence rate of 11.4 per 10,000 delivery hospitalizations [25]. In turn, a similarly designed study conducted in the United States (2013–2016) has reported a rate of 3.8 per 10,000 delivery hospitalizations [26]. Such a wide range of rates is likely related to differences in study design, investigated time periods, and characteristics of the background population. In addition, the lack of a clear, evidence-based, and actionable definition of the condition has likely played a substantial role [7]. The incidence rate identified in our study, 5.5 per 10,000 maternities, is slightly higher than a prior, similarly designed, epidemiological work [23]. This is likely due to the relatively recent time period investigated in our research (2017–2019), since the burden of maternal sepsis has been progressively increasing over time [12,13].

Anemia, delivery by CS, disadvantaged socioeconomic status, black or other minority ethnic group, primiparity, and multiple pregnancy have been frequently cited as independent risk factors for sepsis [22,23,24,25,27,28,29]. However, there is high variability in the reported risk factors, likely due to differences in the adopted definition of sepsis, investigated populations, and study design, and, in some cases, due to the limited sample size. It is worthy of note that sepsis can also occur without any clearly identifiable risk factor [27].

In our cohort, almost 70% of patients had a low education level and approximately one third were foreigners and had a language barrier, all proxies of disadvantaged socioeconomic conditions [30]. Interestingly, one fourth of our women were undergoing antibiotic therapy during the two weeks preceding the diagnosis of sepsis. Although with a lower magnitude, this finding replicates previous data by UKOSS [23], and similarly supports the notion that infections can progress in obstetric patients even following antibiotic therapy, and that close follow-up visits are needed to recognize the deterioration of maternal conditions in a timely fashion.

The case-control analysis among women with postpartum sepsis allowed us to identify the presence of vascular and indwelling bladder catheters as being associated with a more than nine-fold increase in the odds of maternal postpartum sepsis. In turn, nulliparity, prior recurrent infections, pregestational medical conditions, and PROM, previously reported as significant risk factors [23,25], only approached statistical significance. Our findings are in line with recently published evidence, which has highlighted a relevant association of vascular and bladder catheters, particularly if maintained in place for a prolonged period of time, to the risk of infection among pregnant and postpartum women [31,32,33,34]. Efforts towards the promotion of knowledge of adequate aseptic techniques, the potential use of algorithms for correct identification of patients necessitating vascular and bladder catheterization, and the early removal of such catheters should be supported.

Genital tract infection and *E. coli* have been consistently reported as the most common source of infection and causative germs in maternal sepsis cases, respectively [22,23], as we have identified in our analysis. However, differences according to the timing of the onset of maternal sepsis has also been recognized. In line with previous publications, our antepartum cases most frequently displayed urinary tract infections, whereas postpartum cases most frequently displayed genital tract infections [23,35]. This different pattern of infection between antenatal and postnatal cases suggests that thoughtful consideration has to be given to the timing of sepsis when deciding on the antibiotics to prescribe.

This study has also highlighted that respiratory infection is an important cause of maternal sepsis, particularly among pregnant women and those delivered by CS. Interestingly, respiratory infection has been reported as the most frequent source of infection among septic women admitted to the ICU and was associated with longer ICU length of stay and higher absolute mortality risk [28]. This finding emphasizes that knowledge regarding the source of infection in pregnant and recently pregnant women has substantial clinical relevance and should be promoted.

Of note, sepsis patients may lack a clear source of infection or positive cultures [36,37]. We could not identify the source of infection in 18.3% of patients, and 32.4% had negative cultures. Furthermore, 10% of cases had neither a source of infection nor a causative organism identified. Similar findings were reported by the UKOSS study [23], with the inability of identifying the source of infection in 26% of cases, negative cultures in 36.2%, and both conditions in 16.4%.

Maternal mortality reviews, as well as assessment of “near-miss” cases, in high-income countries have repeatedly identified opportunities to improve care among a significant proportion of women who had died or experienced severe morbidity from sepsis [23,24,26,38,39]. A common underlying element in maternal sepsis-related deaths and “near-miss” events is the delay in recognition of the condition, the administration of appropriate treatment, and escalation of care when needed [38]. Maternal sepsis is a time-dependent condition which requires prompt identification and treatment initiation to avoid adverse outcomes. The international Surviving Sepsis Campaign’s guidelines, in line with the “Early Directed Therapy Goal” protocol, have recommended the administration of high-dose intravenous antibiotics within one hour of admission for any woman with suspected sepsis, alongside the early correction of hypotension, if present [40,41,42,43]. In our cohort, 40% of patients started antibiotic treatment late, at more than one hour after sepsis diagnosis. In addition, half of our hypotensive, septic women did not receive volume resuscitation, and vasopressor agents were administered in only 9.5% of cases. Worse outcomes have also been reported in septic women with increased serum lactate levels, an indicator of cellular stress and illness severity, thus suggesting the clinical relevance of their assessment and early correction [44,45]. Such evaluation was missing in 30.6% of our women.

Challenges in the timely recognition and effective treatment of maternal sepsis have been related not only to the lack of a clear definition of the condition [7], but also to similarities between the physiological changes of pregnancy and the symptoms and signs of sepsis [19]. This further highlights the pivotal role of promoting awareness on maternal sepsis among general practitioners, frontline healthcare workers, obstetricians, and midwives in improving quality of care and, ultimately, outcomes [46,47].

Although we did not observe any maternal death due to sepsis, almost one fourth of women experienced a serious adverse event, and 28.3% required ICU admission; in addition, one third of newborns were admitted to the NICU, and 1.8% died in the neonatal period. Similar results were reported by the UKOSS study (ICU and NICU admission rate of 31.2% and 42.3%, respectively) [23]; in turn, the Dutch work identified an ICU admission rate of 79% [21], although a potential overestimation of adverse outcomes has to be considered due to the lack of sepsis as a predefined inclusion criterion.

This study was conducted in a high resource setting with free access to universal healthcare, therefore findings can likely reflect the reality in similar healthcare systems as well as similar populations. Furthermore, similarities of our findings with previously published, methodologically similar studies suggest that they could have generalizable implications for local clinical practice and guideline development. Many clinical messages relate to basic care and can be directed not only to obstetricians and midwives but also to frontline healthcare workers and general practitioners, who may be less aware of the signs and symptoms of maternal sepsis and of the rapidity with which it can progress.

## 5. Strengths and Limitations

The prospective population-based design alongside the nested case-control design, the high participation rate of the maternity units, and the accurate and robust case definition are the main strengths of this study.

The study also has limitations. There was an incomplete coverage of all Italian regions leading to subnational findings, which, however, are unlikely to be biased thanks to the distribution of the participating regions in all the geographical areas of the country and a 95% overall participation rate of the maternity units in these regions. Another potential limitation is the lack of a maternal sepsis code in the ICD-9 Hospital Discharge database, which prevented the ascertainment of completeness of notified cases. Nevertheless, the presence of a trained reference clinician in each participating hospital and the ItOSS monthly checking procedures to monitor case reporting make this limit a minor one. Finally, we were not able to assess the mode of birth as potential risk factor for postpartum sepsis since controls were matched to cases for this variable.

## 6. Conclusions

Although rare, maternal sepsis is a potentially life-threatening complication with a substantial risk of serious adverse events for both the mother and her offspring. Knowledge of its clinical manifestations and high levels of suspicion are essential for prompt diagnosis and treatment.

Coordinated, multi-faceted efforts [5,48] directed to increase awareness of maternal sepsis and its risk factors and management should be promoted, aiming to improve prevention and rates of timely suspected and adequately treated women and, thus, to reduce the burden of infection as a contributing cause of morbidity and mortality.

## Figures and Tables

**Figure 1 microorganisms-11-00105-f001:**
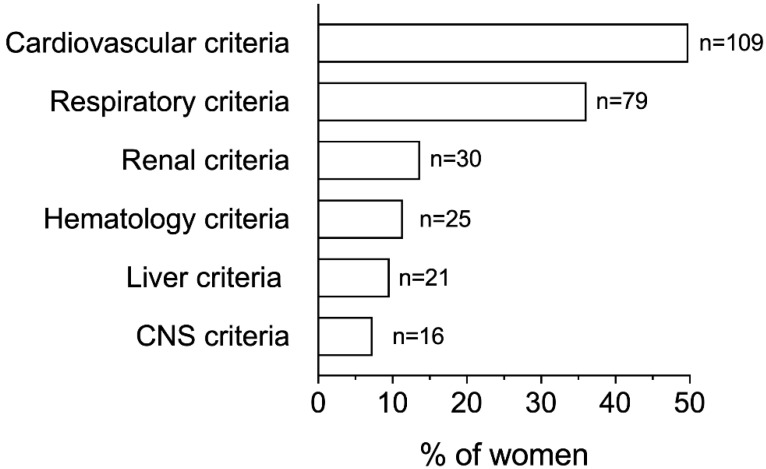
Distribution of ItOSS criteria for diagnosing organ failure among all cases of peripartum sepsis. CNS, Central Nervous System. Each woman could present more than one criterion for diagnosing organ failure. Cardiovascular criteria: systolic blood pressure (SBP) < 90 mmHg or mean arterial pressure (MAP) < 65 mmHg; respiratory criteria: need of oxygen to maintain partial oxygen saturation (SpO2) > 92–93%; renal criteria: creatinine > 1.2 mg/dL or an increase of >0.5 mg/dL and/or diuresis < 0.5 mL/Kg/h for >2 h; hematology criteria: platelets < 100.000/mm^3^ or a decrease of ≥50% compared to usual value in pregnancy; liver criteria: bilirubin > 1.2 mg/dL; CNS criteria: altered state of consciousness.

**Figure 2 microorganisms-11-00105-f002:**
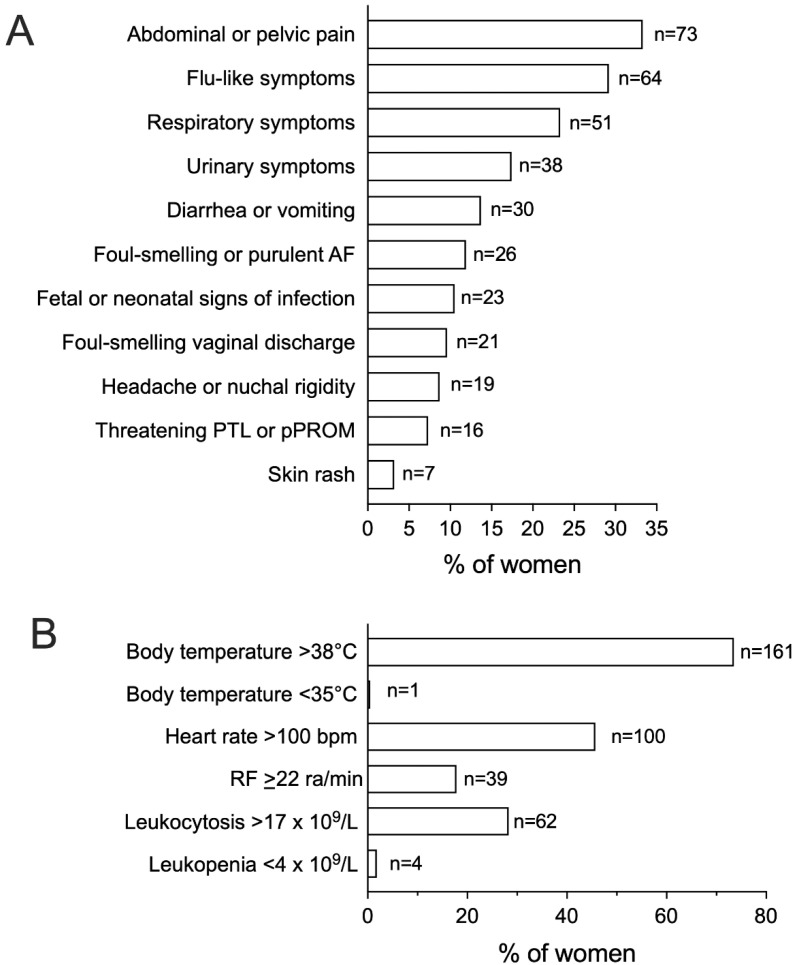
Signs and symptoms of infection, vitals, and laboratory parameters among women with peripartum sepsis. Figure 2 shows the relative frequency of signs and symptoms of infection (**A**) and of presenting vitals and laboratory parameters (**B**) among women with a diagnosis of peripartum sepsis (n = 219). AF, Amniotic Fluid; PTL, Preterm Labor; pPROM, Preterm Premature Rupture of Membranes; RF, Respiratory Frequency; ra, Respiratory Acts.

**Figure 3 microorganisms-11-00105-f003:**
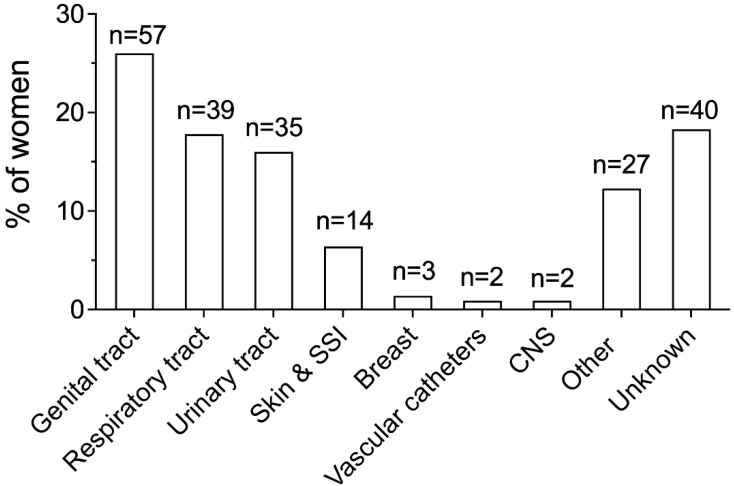
Sources of infection in the overall study population. Genital tract infection included chorioamnionitis (n = 31, 14.2%), endometritis (n = 25, 11.4%), and uterine infection following miscarriage ≥22 weeks’ gestation (n = 1, 0.5%). Urinary tract infection included cystitis (n = 21, 9.6%) and pyelonephritis (n = 14, 6.4%). Breast infection included mastitis and abscesses. CNS infection included meningitis and encephalitis. SSI, surgical site infection; CNS, central nervous system.

**Figure 4 microorganisms-11-00105-f004:**
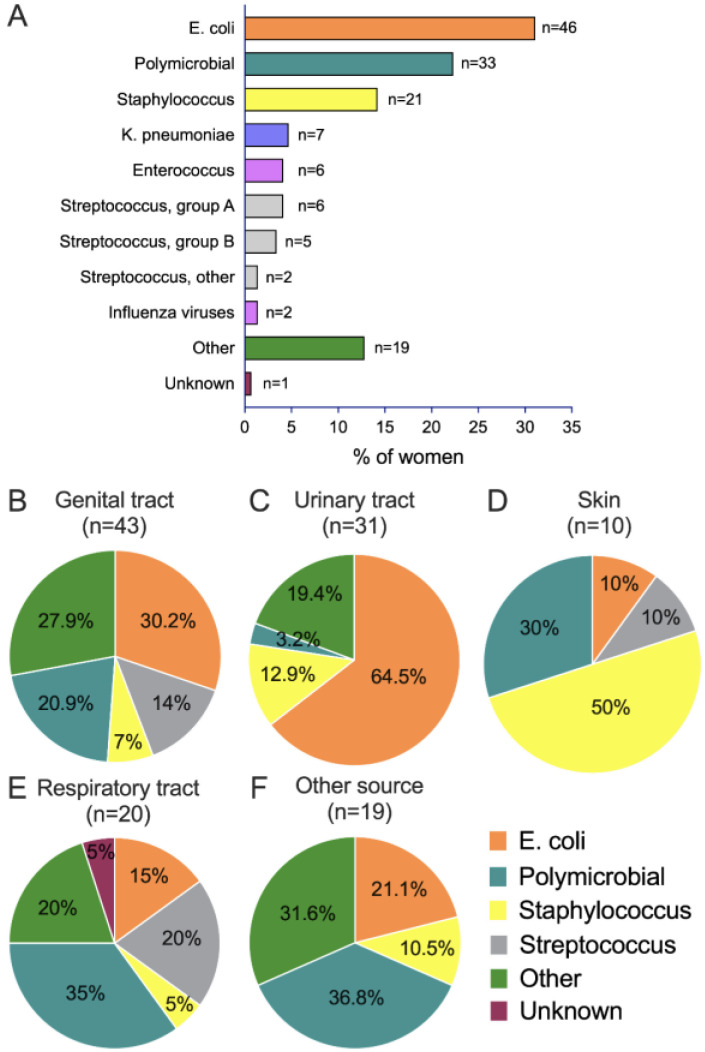
Microorganisms among all septic women with positive culture (**A**) and according to source of infection (**B–F**).

**Figure 5 microorganisms-11-00105-f005:**
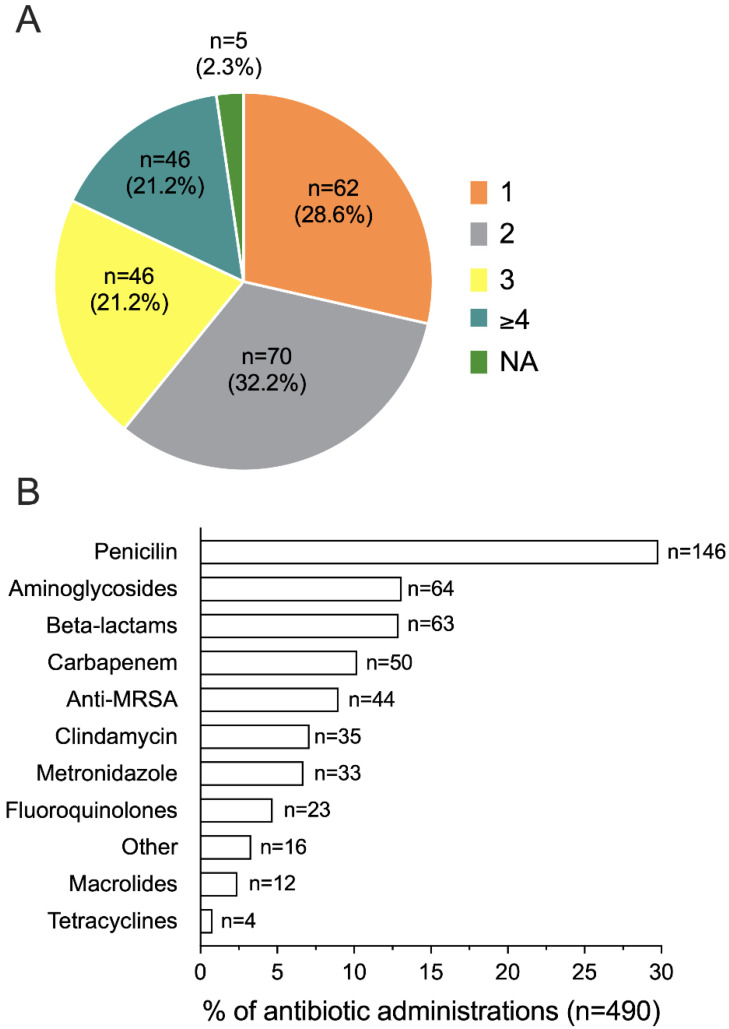
Number (**A**) and type (**B**) of antibiotics administered to septic women (n = 217). NA, not available.

**Table 1 microorganisms-11-00105-t001:** Demographic characteristics and medical and obstetric history of women with peripartum sepsis.

Demographic Characteristics		Overall (N = 219)		Livebirths (N = 212)		Stillbirths	
		N	%	N	%	N	%
Maternal age (years)	<20	7	3.2	7	3.3	0	0.0
	20–35	144	65.8	139	65.6	5	71.4
	≥35	68	31.1	66	31.1	2	28.6
Education level	Low	151	68.9	147	69.3	4	57.1
	High	46	21.0	44	20.8	2	28.6
	Missing	22	10.0	21	9.9	1	14.3
Employment status	Employed	105	47.9	103	48.6	2	28.6
	Unemployed	21	9.6	20	9.4	1	14.3
	Housewife	82	37.4	79	37.3	3	42.9
	Student	3	1.4	3	1.4	0	
	Missing	8	3.7	7	3.3	1	14.3
Citizenship	Italian	139	63.5	135	63.7	4	57.1
	Foreign	74	33.8	71	33.5	3	42.9
	Missing	6	2.7	6	2.8	0	
Ethnicity	White, Italian	133	60.7	129	60.8	4	57.1
	White, Other	38	17.4	36	17.0	2	28.6
	Mixed: White and Black (Caribbean or African)	14	6.4	14	6.6	0	
	Mixed: Other	5	2.3	5	2.4	0	
	Asian	13	5.9	1	6.1	0	
	Black (Caribbean or African or other)	11	5.0	10	4.7	1	14.3
	Other	1	0.5	1	0.5	0	
	Missing	4	1.8	4	1.9	0	
Language barrier (for foreigners)	No	54	73.0	53	74.6	1	33.3
	Yes	20	27.0	18	25.4	2	66.7
Smoking	Never	163	74.4	158	74.5	5	71.4
	Stopped before pregnancy	14	6.4	14	6.6	0	
	Stopped in pregnancy	9	4.1	8	3.8	1	14.3
	Active smoker	19	8.7	18	8.5	1	14.3
	Missing	14	6.4	14	6.6	0	
Pregestational BMI (Kg/m^2^)	<18.5	12	5.5	12	5.7	0	
	18.5–24.9	115	52.5	113	53.3	2	28.6
	25–29.9	46	21.0	43	20.3	3	42.9
	≥30	29	13.2	28	13.2	1	14.3
	Missing	17	7.8	16	7.5	1	14.3
Medical and obstetric history and current pregnancy data		Overall (N = 219)		Livebirths (N = 212)		Stillbirths (N = 7)	
		N	%	N	%	N	%
History of STIs	No	193	88.1	188	88.7	5	71.4
	Yes	12	5.5	11	5.2	1	14.3
	Missing	14	6.4	12	5.7	2	28.6
Prior recurrent infections	No	170	77.6	166	78.3	4	57.1
	Yes	30	13.7	28	13.2	2	28.6
	Missing	19	8.7	18	8.5	1	14.3
Compromised immune system	No	206	94.1	202	95.3	4	57.1
	Yes	3	1.4	2	0.9	1	14.3
	Missing	10	4.6	8	3.8	2	28.6
Diabetes mellitus	No	189	86.3	182	85.8	7	100.0
	Yes	27	12.3	27	12.7	0	
	Missing	3	1.4	3	1.4	0	
Pregestational medical conditions (diabetes excluded)	No	173	79.0	168	79.2	5	71.4
	Yes	46	21.0	44	20.8	2	28.6
Parity	Nulliparas	130	59.4	127	59.9	3	42.9
	Multiparas	89	40.6	85	40.1	4	57.1
Complications in prior pregnancies (for multiparas)	No	56	62.9	55	64.7	1	25.0
	Yes	28	31.5	25	29.4	3	75.0
	Missing	5	5.6	5	5.9	0	
Multiple pregnancy	No	207	94.5	200	94.3	7	100.0
	Yes	11	5.0	11	5.2	0	
	Missing	1	0.5	1	0.5	0	
ART	No	200	91.3	193	91.0	7	100.0
	Yes	18	8.2	18	8.5	0	
	Missing	1	0.5	1	0.5	0	
Amniocentesis	No	213	97.3	207	97.6	6	85.7
	Yes	6	2.7	5	2.4	1	14.3
Complications in current pregnancy	No	74	33.8	74	34.9	0	
	Yes	144	65.8	137	64.6	7	100.0
	Missing	1	0.5	1	0.5	0	
Antibiotic therapy in the 2 weeks before sepsis diagnosis	No	151	68.9	145	68.4	6	85.7
	Yes	57	26.0	57	26.9	0	
	Missing	11	5.0	9	4.2	2	28.6

Education level: high ≥ university degree, low < university degree. Asian ethnicity: India, Pakistan, Bangladesh, China. History of STIs, sexually transmitted infections: HIV, syphilis, N. gonorrhea, C. trachomatis, genital herpes, hepatitis B, and hepatitis C. Prior recurrent infections: recurrent otitis, skin infections, cystitis, pyelonephritis, vaginal infections. Pregestational medical conditions: cardiac disease, endocrine disorders, mental health disorder/psychiatric disease, anemia or other hematological conditions, inflammatory bowel diseases, renal disease, autoimmune conditions, cancer. Complications in prior pregnancies: amniotic fluid embolism, hypertensive disorders of pregnancy, gestational diabetes, hyperemesis gravidarum requiring hospital admission, ovarian hyperstimulation syndrome, small or large for gestational age neonate, placenta previa, placenta abruptio, postpartum hemorrhage requiring blood transfusion, preterm delivery, thromboembolic event, infection requiring hospital admission, stillbirth, surgery in pregnancy, neonatal death. Complications in current pregnancy: amniotic fluid embolism, hypertensive disorders of pregnancy, gestational diabetes, hyperemesis gravidarum requiring hospital admission, ovarian hyperstimulation syndrome, small or large for gestational age neonate, placenta previa, placenta abruptio, postpartum hemorrhage requiring blood transfusion, preterm delivery, thromboembolic event, infection requiring hospital admission, stillbirth, surgery in pregnancy, neonatal death. BMI, Body Mass Index; STI, Sexually Transmitted Infections; ART, Artificial Reproductive Techniques.

**Table 2 microorganisms-11-00105-t002:** Characteristics of the childbirth event.

		Overall (N = 219)		Livebirths (N = 212)		Stillbirths (N = 7)	
		N	%	N	%	N	%
Labor	No	108	49.3	106	50.0	2	28.6
	Yes	107	48.9	103	48.6	4	57.1
	Missing	4	1.8	3	1.4	1	14.3
Induction of labor	No	145	66.2	142	67.0	3	42.9
	Yes	62	28.3	59	27.8	3	42.9
	Missing	12	5.5	11	5.2	1	14.3
Epidural analgesia (for laboring women)	No	46	43.0	43	41.7	3	75.0
	Yes	53	49.5	53	51.5	0	
	Missing	8	7.5	7	6.8	1	25.0
Stained amniotic fluid	No	172	78.5	169	79.7	3	42.9
	Yes, grade 1	7	3.2	7	3.3	0	
	Yes, grade 2 or 3	21	9.6	20	9.4	1	14.3
	Missing	19	8.7	16	7.5	3	42.9
>5 vaginal examinations (for laboring women)	No	81	75.7	77	74.8	4	100.0
	Yes	23	21.5	23	22.3	0	
	Missing	3	6.5	3	2.9	0	
GA at birth	≥37^0/7^ weeks	153	70.0	153	72.2	0	
	32–36^6/7^ weeks	38	17.9	36	17.0	2	28.6
	28–31^6/7^ weeks	18	8.2	17	8.0	1	14.3
	<28^0/7^ weeks	7	3.2	3	1.4	4	57.1
	Missing	3	1.4	3	1.4	0	
Mode of birth	Spontaneous VD	59	26.9	55	25.9	4	57.1
	Operative VD	4	1.8	4	1.9	0	
	Urgent/emergent CS	130	59.4	127	59.9	3	42.9
	Planned CS	26	11.9	26	12.3	0	
Episiotomy	No	54	85.7	50	84.7	4	100.0
	Yes	9	14.3	9	15.3	0	
Type of anesthesia for CS	Loco-regional	128	82.1	127	83.0	1	0.6
	General	28	17.9	26	16.7	2	1.3
Mode of birth of the placenta among VDs	Spontaneous, complete	52	82.5	51	86.4	0	
	Spontaneous, incomplete	1	1.6	1	1.7	1	25.0
	Manual removal, complete	3	4.8	3	5.1	0	0.0
	Missing	7	11.1	4	6.8	3	75.0
Mode of birth of the placenta among CSs	Spontaneous, complete	25	16.0	25	16.3	0	
	Spontaneous, incomplete	2	1.3	2	1.3	0	
	Manual removal, complete	127	81.4	124	81.0	3	7.0
	Missing	2	1.3	2	1.3	0	
PPH > 500 mL	No	176	80.4	170	80.2	6	85.7
	Yes	43	19.6	42	19.8	1	14.3
Peripartum hysterectomy	No	217	99.1	210	99.1	7	100.0
	Yes	2	0.9	2	0.9	0	
Allergic reactions/anaphylaxis	No	218	99.5	211	99.5	7	100.0
	Yes	1	0.5	1	0.5	0	
Anesthesiology complications	No	217	99.1	210	99.1	7	100.0
	Yes	2	0.9	2	0.9	0	

GA, Gestational Age; VD, Vaginal Delivery; CS, Cesarean Section; PPH, Postpartum Hemorrhage.

**Table 3 microorganisms-11-00105-t003:** Sources of infection in septic women according to the timing of sepsis diagnosis and the mode of birth.

	Antepartum (N = 50)		Postpartum (N = 166)		*p*-Value	Vaginal Delivery (N = 63)		Cesarean Section (N = 156)		*p*-Value
	N	%	N	%		N	%	N	%	
Unknown	8	16.0	31	18.7	0.834	14	22.2	26	16.7	0.340
Genital tract Chorioamnionitis Endometritis Uterine infection	7 7 0 0	14.0 14.0	50 24 25 1	30.1 14.5 15.1 0.6	0.028	18 5 12 1	28.6 7.9 19.0 1.6	39 26 13 0	25.0 16.7 8.3	0.612
Urinary tract Cystitis Pyelonephritis	13 7 6	26.0 14.0 12.0	22 14 8	13.3 8.4 4.8	0.045	17 9 8	27.0 14.3 12.7	18 12 6	11.5 7.7 3.8	0.008
Respiratory tract	13	26.0	26	15.7	0.076	4	6.3	35	22.4	0.006
Breast	0		3	1.8	1.000	2	3.2	1	0.6	0.199
Skin & SSI	0		13	7.8	0.042	2	3.2	12	7.7	0.360
CNS	1	2.0	1	0.6	0.410	0		2	1.3	1.000
Vascular catheters	0		2	1.2	1.000	0		2	1.3	1.000
Other	8	16.0	18	10.8	0.328	6	9.5	21	13.5	0.502

Uterine infection: post miscarriage ≥ 22 weeks. Breast infection: mastitis and abscesses. CNS infection: meningitis and encephalitis. SSI, surgical site infection; CNS, central nervous system.

**Table 4 microorganisms-11-00105-t004:** Nested case-control study for women with postpartum sepsis.

		Cases (N = 150)		Controls (N = 297)		Crude OR			Adjusted OR		
		N	%	N	%	OR	95% CI		aOR	95% CI	
**Maternal age (years)**	<35	103	68.7	167	56.2	1.00			1.00		
	≥35	47	31.3	103	34.7	0.58	0.38	0.88	0.80	0.42	1.54
**Citizenship ^a^**	Italian	96	64.0	223	75.1	1.00			1.00		
	Foreign	52	34.7	72	24.2	1.72	1.11	2.68	1.39	0.67	2.88
**Education ^a^**	Low	40	26.7	68	22.9	1.19	0.73	1.94	1.16	0.51	2.64
	High	101	67.3	199	67.0	1.00			1.00		
**Parity**	Nulliparas	88	58.7	132	44.4	1.80	1.21	2.69	1.76	0.94	3.26
	Multiparas	62	41.3	165	55.6	1.00			1.00		
**BMI ^a^**	<18.5	9	6.0	18	6.1	1.00	0.42	2.43	1.24	0.34	4.57
	18.5–24.9	82	54.7	165	55.6	1.00			1.00		
	25–29.9	32	21.3	61	20.5	1.05	0.64	1.73	1.26	0.57	2.79
	≥30	19	12.7	32	10.8	1.16	0.63	2.15	0.78	0.27	2.23
**History of STIs ^a^**	No	289	192.7	142	47.8	1.00			1.00		
	Yes	7	4.7	8	2.7	2.46	0.84	7.17	2.70	0.50	14.44
**Prior recurrent infections ^a^**	No	122	81.3	274	92.3	1.00			1.00		
	Yes	19	12.7	15	5.1	3.08	1.46	6.47	2.68	0.99	7.27
**Pregestational medical conditions**	No	89	59.3	226	76.1	1.00			1.00		
	Yes	61	40.7	71	23.9	2.22	1.44	3.43	1.93	1.00	3.69
**Medical and obstetric history-associated risk factors**											
**Amniocentesis**	No	144	96.0	292	98.3	1.00			1.00		
	Yes	6	4.0	5	1.7	2.51	0.69	9.09	3.43	0.59	19.83
**PROM**	No	129	86.0	282	94.9	1.00			1.00		
	Yes	21	14.0	15	5.1	3.18	1.55	6.51	3.08	0.99	9.53
**Complications in current pregnancy ^b^**	No	105	70.0	260	87.5	1.00			1.00		
	Yes	45	30.0	37	12.5	3.52	2.02	6.12	1.93	0.80	4.64
**Childbirth complications**	No	128	85.3	283	95.3	1.00			1.00		
	Yes	22	14.7	14	4.7	3.87	1.81	8.25	1.59	0.50	5.07
**PPH > 500 mL**	No	117	78.0	265	89.2	1.00			1.00		
	Yes	33	22.0	32	10.8	2.40	1.38	4.18	2.04	0.89	4.69
**Vascular catheters**	No	76	50.7	235	79.1	1.00			1.00		
	Yes	74	49.3	62	20.9	40.84	9.89	168.61	9.02	1.53	53.09
**Intermittent bladder catheterization**	No	138	92.0	291	98.0	1.00			1.00		
	Yes	12	8.0	6	2.0	10.00	2.19	45.64	5.22	0.56	48.90
**Indwelling bladder catheterization**	No	76	50.7	237	79.8	1.00			1.00		
	Yes	74	49.3	60	20.2	41.84	10.14	172.6	9.86	2.00	48.62
**>5 vaginal explorations ^a^**	No	109	72.7	229	77.1	1.00			1.00		
	Yes	33	22.0	35	11.8	2.16	1.20	3.88	1.78	0.79	4.00

^a^ Percentages and odds-ratios calculated including “unknown information” in the analysis. **^b^** Gestational complications (% in cases and controls): anemia (10.9–3.3%); severe infection (2.7–0.3%); respiratory distress syndrome prophylaxis (11.4–4.4%); threatening preterm labor (7.1–3.6%); vagino-rectal swab and/or urine culture positive for GBS (7.6–3.1%). STIs, sexually transmitted infections: HIV, syphilis, N. gonorrhea, C. trachomatis, genital herpes, hepatitis B, and hepatitis C. Prior recurrent infections: recurrent otitis, skin infections, cystitis, pyelonephritis, vaginal infections. Pregestational medical conditions: cardiac disease, diabetes, endocrine disorders, mental health disorder/psychiatric disease, anemia or other hematological conditions, inflammatory bowel diseases, renal disease, autoimmune conditions, cancer. Complications in current pregnancy: amniotic fluid embolism, hypertensive disorders of pregnancy, gestational diabetes, hyperemesis gravidarum requiring hospital admission, ovarian hyperstimulation syndrome, small or large for gestational age neonate, placenta previa, placenta abruptio, postpartum hemorrhage requiring blood transfusion, preterm delivery, thromboembolic event, infection requiring hospital admission, stillbirth, surgery in pregnancy, neonatal death. Childbirth complications: hysterectomy, allergic reactions/anaphylaxis, anesthesiology complications. BMI, Body Mass Index; PROM, premature rupture of membranes; PPH, postpartum hemorrhage.

## Data Availability

Data presented in this study are not publicly available due to ethical standards and legal requirements. Data are available from the Ethics Committee of the Istituto Superiore di Sanità. Italian National Institute of Health (contact via email: segreteria.comitatoetico@iss.it) for researchers who meet the criteria for access to confidential data.

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
