# Peer review of "Maternal Sepsis in Italy: A Prospective, Population-Based Cohort and Nested Case-Control Study"

_microorganisms, 2022, doi:10.3390/microorganisms11010105_

Round 1
Reviewer 1 Report
This is a prospective population-based study on maternal sepsis occurring ≧ 22 weeks’ gestation. The authors conclude that maternal sepsis is a potentially life-threatening complication with a substantial risk of serious adverse events for both the mother and her offspring.
There are many problems regarding to this article.
1.Many patients with maternal sepsis are lack of clear source of infection or positive blood culture. Usually sepsis is diagnosed with a positive blood culture.
2.Many patients with maternal sepsis are lack of laboratory data to support the diagnosis of sepsis such as complete blood count (CBC) with differential and platelet count, C-reactive protein (CRP) or procalcitonin.
3.Many causative organisms are cultured from genital tract, respiratory tract and the urinary tract not from blood. These patients may have genital tract infection, respiratory tract infection or urinary tract infection but not sepsis.
4.Escherichia coli (E.Coli) is the most frequently isolated pathogen, polymicrobial infections are the second most common type of infection. Usually polymicrobial infections are not the causative organisms of the patients with sepsis.
5.Staphylococcus is the third most common type of infection. What type of Staphylococcus is the causative organism? Is it Staphylococcus aureus or Staphylococcus capitis or coagulase negative Staphylococcus?
6.There are 44 patients use anti-MRSA to treat maternal sepsis. What is anti-MRSA? Is it vancomycin or teicoplanin or other drug? There are 64 patients use aminoglycosides, 63 patients use beta-lactams and 50 patients use carbapenem. What kind of aminoglycosides, beta-lactams and carbapenem do you use?
Author Response
This is a prospective population-based study on maternal sepsis occurring ≧ 22 weeks’ gestation. The authors conclude that maternal sepsis is a potentially life-threatening complication with a substantial risk of serious adverse events for both the mother and her offspring.
There are many problems regarding to this article.
- Many patients with maternal sepsis are lack of clear source of infection or positive blood culture. Usually sepsis is diagnosed with a positive blood culture.
- Many patients with maternal sepsis are lack of laboratory data to support the diagnosis of sepsis such as complete blood count (CBC) with differential and platelet count, C-reactive protein (CRP) or procalcitonin.
- Many causative organisms are cultured from genital tract, respiratory tract and the urinary tract not from blood. These patients may have genital tract infection, respiratory tract infection or urinary tract infection but not sepsis.
Answer
We thank the referee for the points raised (n. 1, 2, and 3).
The lack of a clear, evidence-based, and actionable definition of maternal sepsis has challenged the possibility of adequately assess the true burden associated to this condition (Bonet M. et al., Reprod Health 2018). For this reason, an expert consultation has been convened by the WHO in 2016 for developing a new, shared definition of maternal sepsis, which is as follows: a life-threatening condition determined by organ failure resulting from infection during pregnancy, childbirth, or up to 42 days post-abortion or postpartum (WHO, Lancet Glob Health 2017). This definition reflects the thinking embedded in the 2016 Third International Consensus Definitions for Sepsis and Septic Shock (SEPSIS-3) (Singer M. et al., Jama 2016), and it has been employed in the current study. Additionally, since the parameters for sepsis diagnosis developed during the SEPSIS-3 consensus are limited by a low positive predictive value and the lack of validation in the obstetric population, the ItOSS has convened a multidisciplinary panel of experts and defined a set of criteria for diagnosis of organ failure and infection, which were employed in the current study (Donati S., https://www.epicentro.iss.it/itoss/fad-sepsi 2018; https://www.epicentro.iss.it/en/itoss/severe-maternal-morbidity). These criteria are detailed in the ‘Materials and Methods’ section, line 135-149. Of note, a positive blood culture as well as a CBC with differential or increased CRP or procalcitonin levels are not required to suspect or confirm a diagnosis of maternal sepsis. Importantly, identification of suspected or confirmed infection alongside signs or symptoms of organ failure, as well as the other way around, has to lead the clinician to a diagnosis of suspected or confirmed maternal sepsis. Distribution of criteria for diagnosing organ failure among our study population is shown in Figure 1, whereas Figure 2 displays distribution of parameters for diagnosing infection.
- Escherichia coli (E.Coli) is the most frequently isolated pathogen, polymicrobial infections are the second most common type of infection. Usually polymicrobial infections are not the causative organisms of the patients with sepsis.
Answer
We thank the referee for this comment.
Maternal sepsis, particularly in cases with genital tract infection as source of infection, can be sustained by polymicrobial infection (Duff P., Obstet Gynecol 1986; Maharaj D., Obstet Gynecol Surv. 2007; Rosene K. et al., J Infect Dis. 1986). Our findings show that genital tract infection was the most common source of infection in our study population (n=57, 26%), and this was more common in postpartum compared to antepartum women. Of note, postpartum women were the majority in our study population (n=166, 75.8%). A similarly designed study conducted in the United Kingdom reported a 5.2% cases of maternal sepsis sustained by polymicrobial infection (Acosta C.D. et al., Plos Med 2014).
- Staphylococcus is the third most common type of infection. What type of Staphylococcus is the causative organism? Is it Staphylococcus aureus or Staphylococcus capitis or coagulase negative Staphylococcus?
Answer
There were different types of Staphylococcus identified in women included in the current study: Staphylococcus aureus, methicillin resistant Staphylococcus aureus (MRSA), Staphylococcus coagulase negative, Staphylococcus capitis, Staphylococcus epidermidis, Staphylococcus hominis, Staphylococcus lugdunensis, Staphylococcus warneri.
This has now been specified in the ‘Results’ section, Line 355-357 (caption of Figure 4).
- There are 44 patients use anti-MRSA to treat maternal sepsis. What is anti-MRSA? Is it vancomycin or teicoplanin or other drug? There are 64 patients use aminoglycosides, 63 patients use beta-lactams and 50 patients use carbapenem. What kind of aminoglycosides, beta-lactams and carbapenem do you use?
Answer
We thank the referee for this observation.
Unfortunately, data regarding the specific type of antibiotic administered to septic women were not part of the anonymized data collection form used for this study. Reference clinicians in each participating maternity units were required to provide information only on the class of antibiotics used.
Additional references
- Duff P. Pathophysiology and management of postcesarean endomyometritis. Obstet Gynecol 1986;67:269–276
- Maharaj D. Puerperal pyrexia: a review. Part I.Obstet Gynecol Surv. 2007;62(6):393.
- Rosene K, Eschenbach DA, Tompkins LS, Kenny GE, Watkins H. J Infect Dis. 1986;153(6):1028.
Reviewer 2 Report
Dear authors,
This is a well written article regarding a multicenter prospective study on maternal sepsis. I only have some minor comments to make.
In line 25 the abbreviation ItOSS must be explained.
In the Abstract you mention that "a prospective population-based study on maternal sepsis", but you do not mention the aim of this study; "on maternal sepsis" is too general.
In line 183 "nine were missing" should change to "nine had missing".
In line 500 you mention that "findings are likely generalizable to similar healthcare systems". I believe that someone cannot "generalize" because the sample is convenient. This does not mean that the results do not reflect the reality in similar healthcare systems, but with similar populations as well. In order to generalize, the sample must be representative of a prior specified population.
In lines 512-513 you mention "unlikely to be biased thanks to the distribution of the participating regions in all the geographical areas of the country". The distribution is in all geographical areas, but there is no mention in your methodology about clusters, rural and urban areas, percents of sampling regarding each area to reflect analogically the population in these areas.
Author Response
Dear authors,
This is a well written article regarding a multicenter prospective study on maternal sepsis. I only have some minor comments to make.
- In line 25 the abbreviation ItOSS must be explained.
Answer
This has now been addressed.
- In the Abstract you mention that "a prospective population-based study on maternal sepsis", but you do not mention the aim of this study; "on maternal sepsis" is too general.
Answer
We thank the referee for this comment.
The Abstract has now been amended according to the referee’s suggestions (line 26-30).
- In line 183 "nine were missing" should change to "nine had missing".
Answer
This has now been addressed.
- In line 500 you mention that "findings are likely generalizable to similar healthcare systems". I believe that someone cannot "generalize" because the sample is convenient. This does not mean that the results do not reflect the reality in similar healthcare systems, but with similar populations as well. In order to generalize, the sample must be representative of a prior specified population.
Answer
The sentence has now been modified (line 541-542).
- In lines 512-513 you mention "unlikely to be biased thanks to the distribution of the participating regions in all the geographical areas of the country". The distribution is in all geographical areas, but there is no mention in your methodology about clusters, rural and urban areas, percents of sampling regarding each area to reflect analogically the population in these areas.
Answer
We thank the referee for this observation.
All public and private maternity units in the six participating Italian regions were recruited for the study, and only ten of them did not provide the requested data, thus leading to an overall participation of 95%. We believe that this has allowed us to adequately reflect the general obstetric population in the participating regions. This has now been specified in the ‘Discussion’ session, line 555.
Reviewer 3 Report
A study adequately conducted and a well-written manuscript. Some minor comments:
- there is lot of repetition in Introduction and methods.
- mortality rate ad ICU admission rate are not clearly presented. I would highlight this information
- discussion needs shortening
Author Response
A study adequately conducted and a well-written manuscript. Some minor comments:
- There is lot of repetition in Introduction and methods.
Answer
We thank the referee for this comment. The ‘Introduction’ and ‘Materials and Methods’ sections have now been amended accordingly.
- Mortality rate ad ICU admission rate are not clearly presented. I would highlight this information.
Answer
Data regarding ICU admission are reported in the ‘Results’ section, line 379-382. There were no cases of maternal death among our study population; this is specified in the ‘Results’ section, line 383-384. In addition, these findings are reported in the ‘Abstract’, line 36-37, and in the ‘Discussion’ section, line 533-534.
- Discussion needs shortening.
Answer
This comment has now been addressed and the text shortened.
Round 2
Reviewer 1 Report
The authors have not well answered the reviewer's questions and comments.